# Non-Alcoholic Fatty Liver Disease Is Associated with a Decreased Catalase (CAT) Level, CT Genotypes and the T Allele of the -262 C/T *CAT* Polymorphism

**DOI:** 10.3390/cells12182228

**Published:** 2023-09-07

**Authors:** Marcin Kosmalski, Izabela Szymczak-Pajor, Józef Drzewoski, Agnieszka Śliwińska

**Affiliations:** 1Department of Clinical Pharmacology, Medical University of Lodz, 90-153 Lodz, Poland; 2Department of Nucleic Acid Biochemistry, Medical University of Lodz, 92-213 Lodz, Poland; izabela.szymczak@umed.lodz.pl (I.S.-P.); agnieszka.sliwinska@umed.lodz.pl (A.Ś.); 3Central Teaching Hospital of Medical University of Lodz, 92-213 Lodz, Poland; jozef.drzewoski@umed.lodz.pl

**Keywords:** non-alcoholic fatty liver disease, catalase, -262 C/T *CAT* polymorphism

## Abstract

Background: It is well known that oxidative stress plays an important role in the development of non-alcoholic fatty liver disease (NAFLD). It has been suggested that an insufficient antioxidant defense system composed of antioxidant enzymes, including catalase (CAT) and nonenzymatic molecules, is a key factor triggering oxidative damage in the progression of liver disease. Therefore, the aim of our study was to assess whether the level of CAT and -262 C/T polymorphism in the promoter of *CAT* (rs1001179) are associated with NAFLD. Methods: In total, 281 adults (152/129 female/male, aged 65.61 ± 10.44 years) were included in the study. The patients were assigned to an NAFLD group (*n* = 139) or a group without NAFLD (*n* = 142) based on the results of an ultrasound, the Hepatic Steatosis Index, and the Fatty Liver Index (FLI). CAT levels were determined using an ELISA test, and genomic DNA was extracted via the standard phenol/chloroform-based method and genotyped via RFLP-PCR. Results: The CAT level was decreased in NAFLD patients (*p* < 0.001), and an ROC analysis revealed that a CAT level lower than 473.55 U/L significantly increases the risk of NAFLD. In turn, genotyping showed that the CT genotype and the T allele of -262 C/T *CAT* polymorphism elevate the risk of NAFLD. The diminished CAT level in the NAFLD group correlated with increased FLI, waist circumference and female gender. Conclusion: The obtained results support observations that oxidative damage associated with NAFLD may be the result of a decreased CAT level as a part of the antioxidant defense system.

## 1. Introduction

Nonalcoholic fatty liver disease (NAFLD) is one of the most common liver diseases, and its prevalence continues to increase worldwide. It is estimated that the overall spread of NAFLD is about 25%, and it occurs more often in patients with cardiometabolic risk factors, such as obesity, type 2 diabetes (T2DM), hypertension (HA), dyslipidemia and chronic kidney disease (CKD) [1,2,3]. NAFLD is a spectrum disease characterized by hepatic steatosis when no other causes for secondary hepatic fat accumulation can be identified [4,5]. There is a gradual progression of the disease from non-alcoholic fatty liver (NAFL) to non-alcoholic steatohepatitis (NASH) to cirrhosis to hepatocellular carcinoma (HCC). However, this progressive worsening does not occur in all the patients suffering from NAFLD, and significant heterogeneity in the natural history of NAFLD has been observed. Therefore, the presence of two subtypes of NAFLD according to the rate of the progression of fibrosis, rapid and slow, is postulated. Various factors, like T2DM, obesity, old age, and a higher degree of baseline abnormality were identified as possible risk factors for disease progression. Additionally, an older age, low aspartate aminotransferase (AST)/alanine aminotransferase (ALT) ratio, co-morbidities like T2DM or HA, and genetic polymorphisms are probable risk factors for rapid progression [6]. It should be emphasized that the presence of NAFLD is associated not only with the risk of hepatic complications associated with disease progression but also with cardiometabolic diseases such as atrial fibrillation, acute coronary syndrome and ischemic stroke [7]. Of particular importance is the frequent coexistence of NAFLD and T2DM. Although the etiopathogenesis of this coexistence is not fully understood, it seems that liver insulin resistance (IR) and oxidative stress play pivotal roles [8]. It is worth mentioning that a bidirectional relationship between NAFLD and T2DM has been suggested, and the clinical manifestation of IR may begin with a diagnosis of NAFLD, T2DM, or both. However, it is also worth emphasizing that not all patients suffering from NAFLD are at risk of developing T2DM [9].

The major feature of NAFLD is the direct accumulation of fat in the liver, resulting from an imbalance between fatty acids supplied to the organ (originating from the diet, de novo lipogenesis (DNL), and the lipolysis of adipose tissue), lipid synthesis and oxidation as well as TGs transported from the liver as very-low-density lipoproteins (VLDLs) [10]. Recent data demonstrate the involvement of hepatocyte senescence in the development of NAFLD. As the main source of energy and reactive oxygen species (ROS)-producing organelles, mitochondria play a central role in accelerated senescence and the development of NAFLD [11]. Mitochondria are involved in a range of physiological processes, including the production of adenosine triphosphate (ATP), the storage of calcium, the release of free radicals, the oxidation of free fatty acids and the synthesis of cholesterol. Mitochondrial dysfunction has been widely recognized as an important factor in the occurrence and development of NAFLD induced by a high-fat diet (HFD). An HFD can lead to the abnormal accumulation of triglycerides (TGs) and the imbalance of liver mitochondrial function. Liver mitochondrial dysfunction, represented by a decrease in energy production and an impaired redox balance, plays a central role in development of the first and second stages of NAFLD. Mitochondrial dysfunction increases ROS, oxidative stress, and defective bioenergy substances, resulting in the accumulation of fat in the liver and liver injury, which may promote the progression of liver disease from NAFLD to NASH through the mechanisms of liver inflammation, necrosis and fibrosis [12]. Many studies demonstrated that oxidative stress markers (such as ROS and the levels of malonylodialdehyde and 8-oxoguanosine) are elevated in patients suffering from NAFLD [13,14,15].

Antioxidant defense is a part of balanced system that neutralizes excess levels of ROS via enzymatic and non-enzymatic antioxidant defense actions. The primary antioxidant system includes various molecules, such as catalase (CAT), superoxide dismutase and glutathione peroxidase [16]. CAT is present in almost all aerobic organisms and breaks down two hydrogen peroxide molecules into one molecule of oxygen and two molecules of water in a two-step reaction. It has also been reported that CAT is an important enzyme implicated in mutagenesis and inflammatory conditions as well as during the suppression of apoptosis, which are all known to be associated with oxidative stress conditions. It has been postulated that in the liver, CAT may confer cellular protection by degrading hydrogen peroxide into water and oxygen [17]. The regulation of *CAT* gene expression determines CAT levels and involves various mechanisms, including the hypermethylation of CpG islands in the CAT promoter, peroxisome proliferator-activated receptor γ, tumor necrosis factor α and p53 protein. The transcription of the *CAT* gene is also affected by multiple polymorphisms in the gene, including -262 C/T polymorphisms (rs1001179). The latter polymorphism in the promoter region was found to influence the CAT level. In addition to genetic factors, the CAT level can be affected by age, seasonality, physical activity or the number of chemical compounds [18,19]. Alterations in the expression of the CAT gene and the CAT protein level have been reported in a wide variety of diseases, and polymorphisms in the CAT gene have been shown to be associated with various pathophysiological conditions, such as IR, T2DM, type 1 diabetes mellitus (T1DM), gestational diabetes, impaired glucose tolerance, HA, Alzheimer’s disease, breast cancer and therapy, asthma, osteonecrosis, elderly re-nutrition, and bone mineral density [19,20,21,22,23,24,25,26,27,28]. Some studies indicate a relationship between selected *CAT* polymorphisms and liver damage caused by chemical compounds [29], alcohol [30], medications such as valproic acid [31], hepatitis C and B viruses [32,33], as well as the presence of NASH [34] and HCC [35].

Human studies examining the expression or activity of CAT in the course of fatty liver disease have demonstrated conflicting results. Some studies report, that this antioxidant enzyme activity is decreased as fatty liver worsens and the defense mechanism against oxidative stress in the cytosol and mitochondria is damaged [36,37]. On the other hand, some studies have shown that as the fatty liver is aggravated, the CAT activity increases [38,39]. The above suggests that excessive fatty acids supplied by the consumption of a high-fat diet increase fatty acid oxidation and thus catalase activity at the initial stage of fatty liver development [40]. Moreover, the publications addressing the relationship between the polymorphism of CAT, which could be associated with altered CAT level and the risk of NAFLD, are scarce. Therefore, the aim of our study was to assess whether the CAT level and -262 C/T *CAT* polymorphism may be related to NAFLD. We also checked the diagnostic value of CAT level with respect to NAFLD.

## 2. Materials and Methods

### 2.1. Characteristics of Patients

In total, 281 adult patients aged 65.61 ± 10.44 years on the average (including 152 women and 129 men) who were hospitalized between January 2016 and April 2016 for various internal diseases at the Department of Internal Medicine, Diabetology and Clinical Pharmacology at the Medical University of Lodz were included in this observational, case–control study. The study was performed under the guidelines of the Helsinki Declaration for human research and approved by the Medical University of Lodz Committee on the Ethics of Research in Human Experimentation (approval number RNN/210/08/KE).

Prior to the enrolment, each participant provided a written consent to participate in the study. Exclusion criteria included the presence of secondary causes of excessive fatty liver infiltration (ethanol consumption more than 14 g/day; liver diseases; medications, including chemotherapy, combined antiretroviral therapy, amiodarone, methotrexate, tamoxifen, corticosteroids, tetracyclines, valproic acids, amphetamines, and acetylsalicylic acid); genetic causes, such as haemochromatosis, alpha-1 antitrypsin deficiency, Wilson’s disease, congenital lipodystrophy, abetalipoproteinaemia, hypobetalipoproteinaemia, familial hyperlipidaemia, lysosomal acid lipase deficiency, glycogen storage diseases, hereditary fructose intolerance, urea cycle disorders, and citrin deficiency; environmental causes, including lead, arsenic, mercury, cadmium, herbicides, pesticides, polychlorinated biphenyls, and chloroalkenes; nutritional/gastroenterological causes, such as severe surgical weight loss, starvation, malnutrition, total parenteral nutrition, microbiome changes, celiac disease, pancreatectomy, short bowel syndrome; and other causes, such as hypothyroidism, polycystic ovary syndrome, hypothalamic or pituitary dysfunction, growth hormone deficiency, HELLP (hemolysis, elevated liver enzymes and low platelets), Amanita phalloides mushrooms, phosphorous, petrochemicals and Bacillus cereus toxin poisoning), prediabetes and T1DM; gestational or other than T2DM; diseases that may affect muscle metabolism, such as glycogen metabolism disorders; lipid metabolism disorders and mitochondrial myopathies; severe kidney or liver dysfunction (including patients who had had an organ transplant); musculoskeletal damage; or a surgery undergone during the last 6 months, pregnancy and severe infections. The enrolled patients had their medical history taken and underwent a physical examination. The necessary diagnostic tests for glucose metabolism abnormalities (according to ADA) or the presence of secondary causes of fatty liver were performed. Blood pressure (BP, including systolic blood pressure (SBP) and diastolic blood pressure (DBP)), anthropometric measurements (body weight, height, waist (WC) and hip circumference (HC)) were performed and used to calculate the body mass index (BMI) and waist–hip ratio (WHR). Next, blood samples were taken to determine fasting plasma glucose (FPG); glycated hemoglobin level (HbA1c); total cholesterol (T-CH); LDL cholesterol (LDL-CH); HDL cholesterol (HDL-CH); triglycerides (TG); total bilirubin; uric acid; urea; creatinine concentrations; liver enzymes, including alanine aminotransferase (ALT); asparagine (AST); and gamma-glutamyltransferase (GGTP) activity. Postprandial plasma glucose (PPG) was measured two hours after standardized breakfast (equivalent to approximately ~20% of their total energy requirement). These markers were measured by standard laboratory methods. The estimated glomerular filtration rate (eGFR) was calculated based on the Modification of Diet in Renal Disease (MDRD) equations. An ultrasound examination (US) was also performed by an experienced radiologist to assess the presence of hepatic steatosis. Hepatic steatosis was assessed based on observations, including (1) increased liver echogenicity compared to the renal cortex; (2) decreased conspicuity of hepatic vasculature; (3) presence of focal fat sparing; and (4) decreased ability to visualize the diaphragm and deeper liver parenchyma. We also used standard questionnaire to include patients with all factors associated with NAFLD and calculated the Hepatic Steatosis Index (HSI) and Fatty Liver index (FLI) to confirm the presence of fatty liver by ultrasound examination.

HSI was calculated with the following formula:

8 × ALT/AST + BMI (+2 if T2DM yes, +2 if female), where BMI is body weight (kg)/height squared (m^2^). HSI ≥ 36, represents the very high possibility of the presence of NAFLD [41].

FLI was calculated with the following formula:

(e0.953 × loge (TG) + 0.139 × BMI + 0.718 × loge (GGTP) + 0.053 × WC − 15.745)/(1 + e0.953 × loge (TG) + 0.139 × BMI + 0.718 × loge (GGTP) + 0.053 × WC − 15.745) × 100. FLI ≥ 60 indicates that NAFLD is present [42].

Patients were assigned to the NAFLD group (+NAFLD; *n* = 139) and control group (subjects without NAFLD, −NAFLD; *n* = 142, age and sex matched).

If patients did not show evident signs of fatty liver based on US, HSI and FLI, the diagnosis of NAFLD seemed questionable, and the patient was not enrolled to the study.

### 2.2. Determination of CAT Level

After blood collection, the serum was separated and frozen at −80 °C. Plasma CAT level was determined using an ELISA Kit (USCN Life Sciences Inc., Shah Alam, Malaysia) according to the manufacturer’s protocol. CAT level is expressed as U/L.

### 2.3. Genotype Analysis

Blood samples from all study participants were collected into the tubes containing 50 mmol/l disodium-EDTA. Genomic DNA was extracted from peripheral whole blood with the standard phenol/chloroform-based method. All extracted DNA samples were stored at 4 °C until further analysis. All DNA samples from cases and controls were genotyped by polymerase chain reaction (PCR), followed by restriction fragment length polymorphism (RFLP) analysis. In order to avoid potential contamination, the PCR assays were performed with at least one known DNA genotype (positive control) and one negative control (without DNA template). CAT–262 C/T polymorphism was determined using an antisense primer 5′-AGAGCCTCGCCCCGCCGGACCG-3′ and sense primer 5′-TAAGAGCTGAGAAAGCATAGCT-3′. Polymerase chain reaction was performed in a 50 μL volume with 50 ng of genomic DNA, 100 μm dNTPs, 20 pmol of each primer, 2 mM MgCl2, 1 × PCR buffer with (NH4) 2 SO4 and 2 U Taq polymerase. Amplification was carried out in a TC-512 Thermal Cycler (Techne), and the cycling conditions were: 95 °C for 15 min, 35 cycles of 94 °C for 30 s, 60 °C for 45 s, 72 °C for 30 s, and a final extension at 72 °C for 10 min. Amplicons (185 bp) were digested with 10 U of SmaI (Promega, Southampton, UK) at 37 °C for 16 h and analyzed following electrophoresis in 3% agarose gel stained with ethidium bromide (0.5 μg/mL). The T allele was not digested, giving a 185-bp fragment; the C allele was digested, showing two fragments of 155 and 30 bp (due to limitation of agarose gel in the detection of fragments that are smaller than 50 bp, the 30 bp fragment was invisible). Genotyping was performed blindly with respect to case/control status and repeated twice for all subjects, but no discordant genotype classifications were identified.

### 2.4. Statistical Analysis

Anthropometric and biochemical characteristics as well as CAT level were expressed as median with lower and upper quartiles. Their distribution was not in accordance with the normality determined by the Shapiro–Wilk test. The differences between groups were assessed by the Mann–Whitney U test. A chi square test was performed for sex ratio and T2DM ratio comparisons in the groups. Differences between 3 or more groups were checked using the Kruskal–Wallis test with Dunn’s multiple comparison test. The observed number of cases for each genotype in all groups were compared to the expected number from the Chi2 test according to the Hardy–Weinberg principle. The Chi2 test was also used to determine the differences between the distribution of genotypes in all studied groups. The relationship between CAT level and anthropometric and biochemical parameters was evaluated using a Spearman (non-parametric) correlation coefficient. Linear regression analysis was performed to determine the association between CAT level and the clinical and biochemical parameters in +NAFLD group. The non-parametric variables included in the linear regression analysis have not been logarithmically transformed. All analyses were performed using the GraphPad Prism 8.0 software (San Diego, CA, USA). *p* < 0.05 was considered as statistically significant.

## 3. Results

The anthropometric characteristics and BP of the studied groups (+NAFLD and −NAFLD) are presented in Table 1. The +NAFLD group did not differ from the control group (−NAFLD) in terms of age, sex, SBP and DBP. WC as well as HC, WHR and BMI were higher in the +NAFLD group compared to the −NAFLD group. Table 2 displays the biochemical characteristics between the NAFLD and the control group. We found that the +NAFLD group did not differ in terms of PPG, AST, creatinine, eGFR, TCH, LDL-CH or HDL-CH concentration compared to the control group. As expected, the +NAFLD patients had statistically significantly higher FPG, HbA1c, ALT, TG, HSI, FLI, GGTP, total bilirubin and uric acid concentrations. Interestingly, the number of patients suffering from T2DM was statistically greater in the +NAFLD than in −NAFLD group.

### 3.1. CAT Level Is Decreased in NAFLD Group

As depicted in Figure 1, we found decreased CAT levels in the +NAFLD patients compared to the −NAFLD group (340.48 ± 21.80 U/L vs. 471.99 ± 20.72 U/L; *p <* 0.001).

We performed Spearman correlations to show the relationships between the CAT level and age, and between anthropometric and biochemical parameters in the −NAFLD group and the +NAFLD group, as presented in Appendix A. Moreover, the CAT level in +NAFLD group demonstrated a weak positive correlation with eGFR, TCH and LDL-CH, and was negatively correlated with age. In contrast, in the −NAFLD group, there was a weak positive correlation of the CAT level with WC, WHR and ALT. The other parameters did not correlate significantly with the CAT level in either groups.

### 3.2. CAT Level Lower than 473.55 U/L Increases the Risk of NAFLD

In order to find whether the CAT level may be used in the search for NAFLD, several analyses, including Spearman correlations, linear regression, GLM and further ROC analysis were performed. Table 3 shows the results of the linear regression and correlation analysis between the CAT level and age, anthropometric, and biochemical parameters in the whole study population. Among the analyzed parameters, the CAT level positively correlated only with WHR and negatively with age, HSI and FLI. The results of multivariate stepwise linear regression analyses showed that WHR, TG and GGTP values were most strongly correlated with CAT level.

Table 4 presents the association between the CAT level and the presence of NAFLD (with interactions). We observed that decreased CAT levels in +NAFLD group were associated with increased FLI and WC, as depicted in Model 1. In turn, Model 2 revealed that decreased CAT levels in +NAFLD group is related with female gender.

To explore the potential of CAT level as a diagnostic tool for NAFLD risk, ROC analysis was conducted (Figure 2). The optimal cut-point was determined by means of the Younden index method and was 473.55 U/L (AUC = 0.6335, *p* = 0.0006). Further, patients were divided into two subgroups according to the CAT level cut-off point. The difference in frequency of NAFLD between the subgroups was tested by the Chi square test with Yates correction (*p* = 0.0025). A CAT level higher than 473.55 U/L significantly reduced the risk of NAFLD (OR = 0.3936, 95% CI 0.2192–0.7065, *p* = 0.0025).

### 3.3. CT Genotype and Allele T of -262 C/T Polymorphism of CAT Increases the Risk of NAFLD

The results of the genotyping of -262 C/T *CAT* polymorphism in −NAFLD and +NAFLD groups are shown in Table 5. As one can see, +NAFLD patients present a significantly higher frequency of CT genotype than the control group (−NAFLD) in comparison to wild-type genotype (CC). Similarly, there is a slightly higher frequency of TT genotype, but not significant, in the +NAFLD group in comparison to the −NAFLD group in relation to the CC genotype. Moreover, we found that the CT genotype significantly increases the risk of NAFLD (OR = 1.808; CI = 1.072–3.048; *p* = 0.0256), as well as T allele (OR = 1.692; CI = 1.037–2.759; *p* = 0.0345).

We also presented the CAT levels in the +NAFLD and −NAFLD groups according to the genotypes (CC, CT and TT) and alleles (CC, CT + TT), for which the data are shown in Appendix A. Although not statistically significant, the +NAFLD group had a reduced level of CAT independently of the genotype. Notably, the CC carriers in +NAFLD group had significantly decreased CAT levels in comparison to the carriers of CC in the control group (Appendix A). The analysis, considering alleles C or T, also suggests that CAT levels depended on the presence of NAFLD rather than on the allele type (Appendix A). Thus, the presented results indicate that the CAT level is related to the presence of NAFLD, but not to the CAT–262 C/T genotype. We did not observe any significant changes in the CAT level in CT, TT genotypes’ carriers, as well as T allele carriers with NAFLD and without NAFLD.

We also determined anthropometric, BP and biochemical parameters in +NAFLD and −NAFLD groups according to genotypes (Appendix A). We found statistically significant higher level of WC, HC, BMI and WHR in carriers of the CC genotypes in the +NAFLD group compared to the −NAFLD group. Additionally, the +NAFLD group carrying CT genotype also had higher WC and BMI values than CT genotype carriers without NAFLD. We also found statistically significant higher levels of ALT, GGTP, TG, uric acid, HSI and FLI in CC genotype patients with NAFLD compared to CC genotype patients without NAFLD. Additionally, CT genotype carriers had significantly higher TG, HSI and FLI in the +NAFLD group compared to the control group. Similar results were observed in higher FLI and HSI in TT genotype carriers in +NAFLD vs. −NAFLD (Appendix A). Considering the C and T alleles of -262 C/T *CAT* polymorphism, we found that it had no significant effect on differences in age, anthropometric measurements and BP values in both the +NAFLD group and the control group. However, we found that the presence of T alleles correlated with the presence of higher WC, HC and BMI in the +NAFLD group compared to the −NAFLD group (Appendix A). This observation further suggest that anthropometric parameters are associated with NAFLD but not with -262 C/T *CAT* polymorphism. We also observed that T allele carriers in the +NAFLD group had statistical significant higher FPG, GGTP, total bilirubin, TG, uric acid, HSI and FLI than T allele carriers in the control group. On the other hand, patients with NAFLD and C allele had statistically significantly higher ALT, AST, GGTP, TG, uric acid, HSI and FLI that patients with C allele but without NAFLD. It should be emphasized that there was no relationship between the C and T alleles and anthropometric parameters, laboratory parameters, and BP values in both study groups. The exception was the increased value of FPG among subjects with the C allele (Appendix A).

The results of the correlation analysis focused on the type of genotype CAT level showed a positive correlation with WHR, eGFR, TCH and GGTP, and a negative one with age and HC in the +NAFLD group carrying the CT genotype of -262 C/T *CAT* polymorphism. Additionally, we observed a positive correlation between CAT level and TG in the +NAFLD group carrying the CT genotype of -262 C/T *CAT* polymorphism. A negative correlation with age and weak positive correlation with TCH and HDL-CH were observed in the +NAFLD patients carrying the CC genotype of -262 C/T *CAT* polymorphism. In turn, in the −NAFLD group, the level of CAT presented a moderate positive correlation with ALT in the carriers of the CC genotype of -262 C/T *CAT*. We also found a moderate positive correlation between CAT level and FPG in the carriers of the CT genotype of -262 C/T *CAT* in the −NAFLD group. In the +NAFLD group, the CAT level was weakly negatively correlated to age and weakly positively with TCH and HDL in carriers of the CC genotype of -262 C/T polymorphism. In turn, carriers of the CT genotype of the +NAFLD group presented a moderate negative correlation between the CAT level and age and HC, and a moderate positive correlation between the CAT level and WHR, eGFR, FPG, TCH, TG and GGTP (Appendix A). We also found a positive correlation between the CAT level and WHR and ALT in C allele carriers in the group without NAFLD. In the same group, the carriers of the T allele showed a positive correlation between the CAT level and WHR. In the NAFLD group, the CAT level was positively correlated with TCH and HDL-CH. On the other hand, in the same group, but in patients carrying T allele, the CAT level was positively correlated with WHR, TCH and GGTP and negatively with age, HC and SBP (Appendix A). Thus, the obtained results suggest that CAT level correlates with anthropometric (WC, WHR and BMI) and biochemical (ALT, GGT) parameters as well as HSI and FLI, which are typical of the presence of NAFLD and are characteristic symptoms of this disease.

## 4. Discussion

The role of CAT in the etiopathogenesis of NAFLD is not fully established. The results of animal and human studies are inconclusive. In animal studies, it was found that CAT knockout may play an important role in the development and progression of NAFLD. These studies indicate that endogenous CAT exerts beneficial effects on the protection of liver damage, including lipid accumulation and inflammation, by maintaining the redox balance of the organ from the early stages of high-fat diet (HFD)-induced metabolic stress [43,44,45]. In most cell and animal experiments, the activity or expression of CAT was measured after H_2_O_2_ treatment or steatosis induced by obesity. CAT activity or mRNA expression increases after H_2_O_2_ treatment (by ROS stimulation). Because oxidative stress is initiated during an early stage of steatosis, consistently high CAT activity is observed [46]. In a clinical study, CAT expression or activity in patients suffering from NAFLD was not consistent because the measurements were performed at different stages of NAFLD [40]. It was demonstrated that the elimination of CAT easily causes steatosis in mice by promoting excessive lipid accumulation. This suggests that CAT plays a protective role as an antioxidant in the livers with NASH, given that oxidative stress is an important therapeutic or preventive target for patients with NASH [47].

A study of human subjects showed that CAT expression, which increased mainly in the early stage of NAFLD and decreased in the terminal stage of NASH, differs according to the stage of NAFLD [40]. Peroxisomes are essential organelles for maintaining the homeostasis of lipids and ROS. While oxidative stress-induced endoplasmic reticulum (ER) stress plays an important role in NAFLD, the role of peroxisomes in ROS-mediated ER stress in the development of NAFLD remains elusive. Hwan et al. demonstrated that the inhibition of CAT by 3-aminotriazole in hepatocytes resulted in (i) increased peroxisomal H_2_O_2_ levels, as measured by a peroxisome-targeted H_2_O_2_ probe (HyPer-P); (ii) elevated intracellular ROS; (iii) decreased peroxisomal biogenesis; (iv) activated ER stress; (v) induced lipogenic genes and neutral lipid accumulation; and (vi) suppressed insulin signaling cascade associated with JNK activation. N-acetylcysteine or 4-phenylbutyric acid effectively prevented these alterations [48]. On the contrary, in a study conducted by Świderska et al., 67 patients with NAFLD had significantly higher CAT activity. It should be emphasized that in this study, NAFLD was diagnosed based on the liver steatosis in the US examination, as well as ALT activity. In the group of patients with early NAFLD (patients without fibrosis in elastography and normal ALT activity), similar CAT levels were found in the group of patients with advance NAFLD (with fibrosis and upper ALT activity). It is noteworthy that CAT levels in patients with early and advanced NAFLD were significantly lower those in the non-NAFLD group. It is worth underlining that in the study population, NAFLD patients were characterized by significantly higher values of BMI, ALT and AST. However, there were no significant differences in WHR, GGTP, lipid profile and FPG with and without NAFLD. In addition, patients with NAFLD differed from early NAFLD only in higher ALT and AST activity but had the same GGTP values, anthropometric measurements, lipid profile and glucose concentration. It should be emphasized that the CAT value significantly negatively correlated only with WHR and HDL in the group of patients with NAFLD. This correlation was not found for body weight and TCH [49]. Similar results regarding the relationship between CAT and NAFLD were obtained by Perlemuter et al. Only 16 patients with NAFLD diagnosed on the basis of ALT activity, BMI values and a hepatorenal difference in echo intensities participated in this study and were compared to 16 individuals without NAFLD. It is noteworthy that in this study, patients with NAFLD had significantly higher values of BMI, ALT, AST, TG, FBG and HbA1c than patients without NAFLD [39]. On the other hand, Das et al. revealed that NAFLD patients, compared to patients without liver steatosis had 18.5% reduced CAT activity. That study was also conducted on a small group of 35 patients with NAFLD. It was highlighted that patients with NAFLD showed significantly higher BMI, TCH, LDL-cholesterol, VLDL-cholesterol levels, and ALT and AST activities compared to patients without NAFLD. The patients were broadly classified as part of the NAFLD group on the basis of an oral questionnaire, laboratory investigations, clinical findings and US/CT scan imaging or biopsy [50]. Despite the fact that in the study by Kumar et al. NAFLD patients did not differ in terms of age, sex, BMI values, it was found that they had a significantly lower CAT level from healthy controls [51].

Our results showed that NAFLD patients had significantly lower CAT level than non-NAFLD patients. It should be emphasized that our study included as many as 139 patients with NAFLD. All secondary causes of fatty liver were excluded, and we used not only US examinations, but also HSI and FLI scales for NAFLD diagnosis. In our study, patients with NAFLD did not differ in terms of age, sex, BP values, nor PPG, AST, creatinine, eGFR, TCH, LDL-CH and HDL-CH concentration compared to the non-NAFLD group. However, patients with NAFLD had statistically significant higher anthropometric parameters (WC, HC, BMI and WHR) and metabolic parameters associated with the disease (FPG, HbA1c, ALT, TG, his, FLI, as well as GGTP, total bilirubin and uric acid concentrations). Nevertheless, the number of patients suffering from T2DM was statistically significantly greater in patients with NAFLD than in patients without NAFLD. For the first time we found the cut-off point level (473.55 U/L) of CAT relevant for NAFLD diagnosis. Additionally, the CAT level positively correlated only with WHR and negatively with age, HSI and FLI. The results of multivariate stepwise linear regression analyses showed that WHR, TG and GGTP values were most strongly and significantly correlated with the CAT level. We observed, that decreased CAT levels in patients with NAFLD were associated with increased FLI, WC and female gender. The correlation analysis focused on the type of genotype revealed that such CAT level is correlated with anthropometric (WC, WHR and BMI) and biochemical (ALT and GGT) markers as well as HSI and FLI, which are typical of the presence of NAFLD and are the characteristic symptoms of this disease.

Among numerous polymorphisms of the CAT gene (for example, genetic polymorphisms of -21 A/T (rs7943316), -262 C/T (rs1001179), and -844 C/T (rs769214), all in the promoter region) significantly affect *CAT* expression via the alteration of the recognition sites of transcriptional factors [52]. Moreover, -262 C/T *CAT* polymorphism is in the promoter region of the CAT gene and was found to be associated with altered CAT levels. Komina et al. revealed that CT and TT genotypes were associated with the decreased CAT levels in contrast to CC carriers who had elevated CAT levels in the presence of 0.4 and 0.7 mM H_2_O_2_ [53]. In contrast, it was found that CAT levels were significantly higher in donors carrying the T allele in comparison to donors homozygous for the C allele [54]. The results of our study are inconclusive on this point because we did not obtain statistically significant differences, and regardless of genotype and allele, +NAFLD patients had lower CAT levels than −NAFLD patients. Moreover, our data also suggest that anthropometric parameters were associated with NAFLD, but not with -262 C/T *CAT* polymorphism. We also found statistically significantly higher level of ALT, GGTP, TG, uric acid, HSI and FLI in CC genotype patients with NAFLD compared to CC genotype patients without NAFLD. CT genotype carriers had significantly higher TG, HSI and FLI in the +NAFLD group compared to the control group. T allele carriers in the +NAFLD group had statistically significantly higher FPG, GGTP, total bilirubin, TG, uric acid, HSI and FLI than T allele carriers in the control group. On the other hand, the patients with NAFLD and C allele had statistically significantly higher ALT, AST, GGTP, TG, uric acid, HSI and FLI than patients with C allele but without NAFLD. Taken together, the reported correlations between the genotype and alleles of -262 C/T *CAT* polymorphism and the CAT level, anthropometric, and biochemical parameters in the studied groups suggest that they result from NAFLD rather than this polymorphism.

There is little evidence concerning the relation between -262 C/T *CAT* polymorphism and the risk of NAFLD. Huang et al. found that patients with NAFLD had significantly higher frequency of T alleles (10.7%) than the control group (7.2%) and lower frequency than the NASH group (23.0%). Their study included 56 patients with NAFLD and 126 with NASH (vs. 153 controls), whose progression of hepatic fatty infiltration was assessed based on biopsy. After adjusting for confounders, the CAT mutant T alleles were still the highest independent risk factors for NASH. It is emphasized that there were no statistically significant differences between the gender in ratios of the NASH, NAFLD and control groups. However, the NASH and NAFLD groups had an increased age, BMI, serum ALT and AST levels, and incidence of T2DM and hyperlipidemia compared to the control group. The NASH group had also a higher BMI, ALT and AST compared to the NAFL group [34]. Our study revealed that the CT genotype as well as T allele significantly increases the risk of NAFLD. Additionally, CC carriers in the NAFLD group had significantly decreased CAT level in comparison to carriers of CC in the control group. We did not observe any significant changes in CAT level in CT and TT genotypes (allele T carriers) between patients with NAFLD and without NAFLD. Thus, it seems that the CAT level is related to the presence of NAFLD but not to the genotypes of -262 C/T *CAT*.

Considering the relationship between NAFLD, obesity, IR and T2DM, it is suggested that CAT and its gene polymorphisms may play an important role in the coexistence of these pathologies. Hernández-Guerrero et al. proved that the CAT-262 frequency of the homozygous wild-type genotype in subjects in the control group was 0.72, whereas for the obesity group, it was 0.78. For the mutated carriers (TC + TT), the frequency was (0.28) and (0.22) for the normal weight and the obesity group, respectively. Additionally, they did not find any statistical differences when comparing the frequencies in subjects presenting diabetes or impaired fasting glucose (IFG) and dyslipidemia vs. the participants with obesity but with no diabetes, IFG or dyslipidemia. A significant elevated body fat percentage was observed in subjects who were carriers of the mutated genotype compared to the wild-type genotype in the case of CAT-262. In the same aspect, CAT-262 presented statistical differences when visceral fat was compared between groups of genotypes [55]. On the other hand, Apraci et al. revealed that CAT activity was observed to be significantly lower in the TT genotype compared to CT genotype in the T2DM group. Moreover, CAT (*p* = 0.004 for the CC genotype, *p* = 0.0001 for CT and TT genotypes, respectively) enzyme activity was observed to be significantly lower in the T2DM compared to the control group [56]. Góth et al. further noted that patients with T2DM with the CC or CT genotypes of -262 C/T had decreased CAT and increased glucose and HbA1c levels. T2DM patients who have the TT genotype in -262 C/T may have elevated risk of diabetes complications; these patients had the lowest mean CAT and HDL, as well as the highest glucose, HbA1 and TCH levels [57]. Additionally, there is evidence that CAT polymorphism is associated with diabetes complications, such as distal symmetric polyneuropathy, nephropathy, retinopathy and risk factors for its development [20,25,58,59]. In view of the significant relationship between T2DM and NAFLD [9], and the discussion related to the sequence of development of these pathologies as well as the etiopathogenetic factors involved in this process, it is worth considering the role of CAT, its genotypes and alleles in this process. Our results suggest that CAT, the CT genotype and T allele may be involved in this process

This is a case–control observational study; therefore, some strengths and limitations should be indicated. We made an effort to match controls to cases to avoid bias; however there were more patients with T2DM in the +NAFLD group than in −NAFLD group. In addition, despite the use of a triple tool for diagnosing NAFLD (ultrasounds examination, HSI and FLI), the presence of NAFLD was not confirmed by liver biopsy. The strength of the study is the large number of patients included.

## 5. Conclusions

The results of our study showed reduced level of CAT in the group of patients with NAFLD compared to patients without NAFLD. A CAT level lower than 473.55 U/L was found to increase the risk of NAFLD. In addition, CT genotype and the T allele of -262 C/T *CAT* polymorphism significantly increase the risk of NAFLD.

## Figures and Tables

**Figure 1 cells-12-02228-f001:**
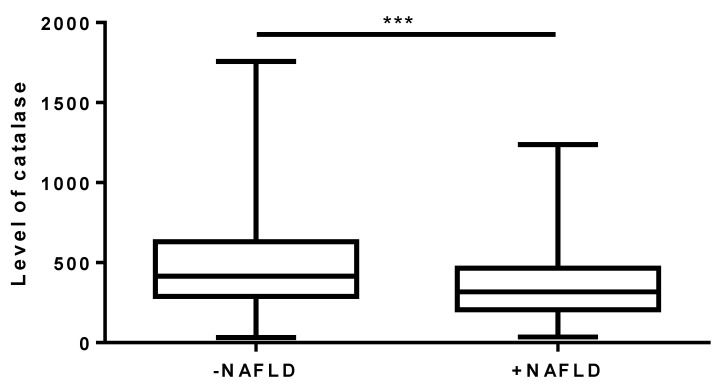
CAT level in the group of patients with NAFLD (+NAFLD, *n* = 139) and without NAFLD (−NAFLD, *n* = 142), measured using ELISA test. The data are expressed as medians with lower and upper quartiles and minimal and maximal values. *** *p* < 0.001 −NAFLD group vs. +NAFLD group.

**Figure 2 cells-12-02228-f002:**
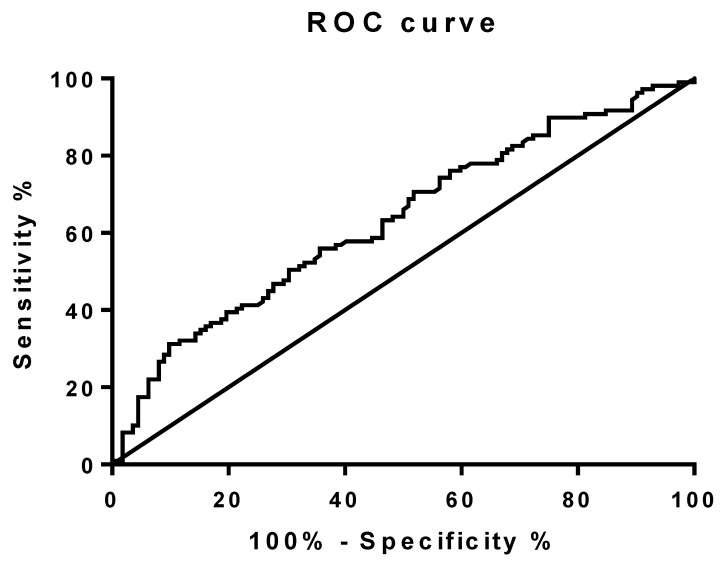
Receiver operating characteristic (ROC) curve of CAT for NAFLD.

**Table 1 cells-12-02228-t001:** Anthropometric characteristics of the study groups, including age, sex, blood pressure and presence of T2DM.

Parameter *	Group 0 −NAFLD (*n* = 142)	Group 1 +NAFLD (*n* = 139)	*p* **
Age (years)	68.50 (57.75; 77.00)	65.00 (57.00; 74.00)	0.0842
WC (cm)	98.50 (90.00; 107.00)	107.00 (101.00; 117.80)	**<0.001**
HC (cm)	106.00 (98.00; 111.00)	110.00 (104.00; 120.00)	**<0.001**
BMI (kg/m^2^)	27.57 (23.82; 19.98)	31.19 (28.05; 34.57)	**<0.001**
WHR	0.9390 (0.8809; 0.9825)	0.9668 (0.9123; 1.023)	**0.01**
SBP (mmHg)	130.00 (120.00; 140.00)	140.00 (125.00; 150.00)	0.0861
DBP (mmHg)	80.00 (70.00; 80.00)	80.00 (70.00; 90.00)	0.3238
Sex (% of F)	53.52	54.68	0.8643

* BMI—body mass index, DBP—diastolic blood pressure, F—female, HC—hip circumference, NAFLD—non-alcoholic fatty liver disease, SBP—systolic blood pressure, WC—waist circumference, WHR—waist-hip ratio. ** *p*-value assessed using the Mann—Whitney U test, except for the sex variables, for which the chi-square test was used. The data are expressed as median (Quartile 1; Quartile 3). The bolded results indicate statistically significant differences.

**Table 2 cells-12-02228-t002:** Biochemical characteristics of the study groups.

Parameter *	Group 0 −NAFLD (*n* = 142)	Group 1 +NAFLD (*n* = 139)	*p* **
FPG (mmol/L)	6.470 (5.018; 10.19)	7.950 (5.580; 10.66)	**0.0311**
PPG (mmol/L)	8.080 (5.938; 14.64)	9.740 (6.365; 14.75)	0.3818
HbA1c (%)	7.595 (6.040; 9.148)	8.460 (6.760; 10.00)	**0.0335**
ALT (U/L)	19.00 (13.00; 29.00)	25.00 (18.00; 36.50)	**0.0003**
AST (U/L)	20.00 (16.00; 28.75)	23.00 (17.00; 34.50)	0.0642
GGTP (U/L)	22.00 (16.36; 45.00)	43.00 (25.61; 75.00)	**<0.0001**
Total bilirubin (μmol/L)	8.500 (6.910; 12.82)	10.55 (7.745; 15.36)	**0.0072**
Creatinine (μmol/L)	73.00 (62.00; 100.5)	76.00 (64.75; 93.25)	0.7111
eGFR (mL/min/L, 73 m^2^)	84.00 (55.50; 104.5)	81.00 (64.00; 101.5)	0.8855
TCH (mmol/L)	4.270 (3.493; 5.175)	4.530 (3.715; 5.400)	0.1049
LDL-CH (mmol/L)	2.600 (1.840; 3.400)	2.685 (1.863; 3.400)	0.9116
HDL-CH (mmol/L)	1.110 (0.9300; 1.390)	1.030 (0.8100; 1.330)	0.2537
TG (mmol/)	1.220 (0.830; 1.610)	1.790 (1.210; 2.685)	**<0.0001**
Uric acid (μmol/L)	276.00 (239.00; 331.00)	352.00 (283.50; 428.00)	**<0.0001**
HSI	31.00 (29.30; 33.00)	43.20 (38.90; 49.38)	**<0.0001**
FLI	32.49 (21.19; 53.04)	85.31 (75.97; 95.99)	**<0.0001**
T2DM (%)	62.68	74.82	**0.0282**

* ALT—alanine aminotransferase, AST—aspartic aminotransferase, FPG—fasting plasma glucose, PPG—postprandial plasma glucose, GGTP—gamma-glutamyltransferase, eGFR—estimated glomerular filtration rate, FLI—Fatty Liver Index, HbA1c—glycated hemoglobin, HDL-CH—HDL cholesterol, HSI—Hepatic Steatosis Index, LDL-CH—LDL cholesterol, NAFLD—non-alcoholic fatty liver disease, T2DM—type 2 diabetes mellitus, TCH—total cholesterol, TG—triglycerides. ** *p*-value assessed using the Mann—Whitney U test, except for the T2DM variables, for which the chi-square test was used. The data are expressed as median (Quartile 1; Quartile 3). The bolded results indicate statistically significant differences.

**Table 3 cells-12-02228-t003:** Spearman correlations and linear regression analysis of CAT level, anthropometric and biochemical parameters in the whole study population (*n* = 281). Only the best CAT-associated variables are shown, as assessed by the stepwise forward multiple regression method.

	Whole Study Population (*n* = 281)			
Parameters *	Rho ***	*p* **	β ± SE ****	*p* **
Age (years)	**−0.1426**	**0.0341**		
BMI (kg/m^2^)	−0.0691	0.3362		
WC (cm)	0.0495	0.4976		
HC (cm)	−0.0720	0.3238		
WHR	**0.1444**	**0.0469**	**0.2472 ± 0.1108 ^#^**	**0.0288**
SBP (mmHg)	−0.1048	0.1210		
DBP (mmHg)	−0.0858	0.2052		
Creatinine (μmol/L)	0.0362	0.5945		
eGFR (mL/min/1.73 m^2^)	0.1240	0.0670		
FPG (mmol/L)	0.0843	0.2205		
PPG (mmol/L)	0.0419	0.5427		
HbA1c (%)	0.0962	0.2246		
Uric acid (μmol/L)	−0.0675	0.4248		
TCH (mmol/L)	0.0927	0.1810		
LDL-CH (mmol/L)	0.1255	0.0715		
HDL-CH (mmol/L)	0.0286	0.6836		
TG (mmol/)	−0.0905	0.1935	**−0.5073 ± 0.2146 ^#^**	**0.0208**
ALT (U/L)	0.1284	0.0614		
AST (U/L)	−0.0214	0.7557		
Total bilirubin (μmol/L)	0.0893	0.1983		
GGTP (U/L)	−0.0872	0.2265	**−0.2730 ± 0.1345 ^#^**	**0.0461**
HSI	**−0.1431**	**0.0494**		
FLI	**−0.1741**	**0.0263**		

* ALT—alanine aminotransferase, AST—aspartic aminotransferase, BMI—body mass index, DBP—diastolic blood pressure, eGFR—estimated glomerular filtration rate, HDL-CH—HDL cholesterol, HSI—Hepatic Steatosis Index, FLI—Fatty Liver Index, FPG—fasting plasma glucose, GGTP- gamma-glutamyltransferase, HbA1c—glycated hemoglobin, HC—hip circumference, LDL-CH—LDL cholesterol, NAFLD—non-alcoholic fatty liver disease, PPG—postprandial plasma glucose, SBP—systolic blood pressure, TCH—total cholesterol, TG—triglycerides, WC—waist circumference, WHR—waist-hip ratio. ** Rho—Spearman rank correlation coefficient. *** *p*-value. β ± SE ****—regression coefficient ± standard error. # variables were log-transformed prior to linear regression analysis. The bolded results indicate statistically significant associations.

**Table 4 cells-12-02228-t004:** Results of the general linear model (GLM) for catalase concentration and NAFLD status (with interactions). The models with a sigma-restricted parameterization, and effective hypothesis decomposition, ANOVA and ANCOVA test.

	Effect *	SS **	DF ***	MS ****	F *****	*p* ******
Model 1	Intercept	56,817	1	56,817.0	1.085118	0.299167
	HSI	30,144	1	30,143.5	0.575695	0.449149
	FLI	**486,068**	**1**	**486,068.4**	**9.283160**	**0.002717**
	Age	66,267	1	66,266.6	1.265589	0.262324
	BMI	18	1	18.3	0.000349	0.985109
	**WC**	**322,560**	**1**	**322,559.6**	**6.160394**	**0.014124**
	Error	8,168,196	156	52,360.2	1.085118	0.299167
Model 2	Intercept	1,679,213	1	1,679,213	32.75987	0.000000
	HSI	19,216	1	19,216	0.37488	0.541265
	FLI	4095	1	4095	0.07988	0.777837
	**NAFLD**	**307,417**	**1**	**307,417**	**5.99741**	**0.015456**
	TG	74,163	1	74,163	1.44685	0.230892
	GGTP	39,665	1	39,665	0.77383	0.380415
	**Sex (Woman)**	**558,268**	**1**	**558,268**	**10.89128**	**0.001202**
	Error	7,842,512	153	51,258		

* BMI—body mass index, FLI—Fatty Liver Index GGTP—gamma-glutamyltransferase, HSI—Hepatic Steatosis Index, NAFLD—non-alcoholic fatty liver disease. TG—triglycerides WC—waist circumference. ** SS—sums of squares. *** DF—degrees of freedom. **** MS—mean square. ***** F—F test statistic. ****** *p*-value. The bolded results indicate statistically significant associations.

**Table 5 cells-12-02228-t005:** The genotype, allele frequency and odd ratios (OR) of the -262 C/T *CAT* polymorphism in patients with NAFLD (+NAFLD) and without NAFLD (−NAFLD).

Genotype CAT	−NAFLD * (*n* = 142)		+NAFLD * (*n* = 139)		OR (95% CI)	*p* **
	N	%	N	%		
CC	98	69	79	57	Reference	-
CT	35	25	51	37	1.808 (1.072–3.048)	**0.0256**
TT	9	6	9	6	1.241 (0.470–3.274)	0.6629
CT + TT	44	31	60	43	1.692 (1.037–2.759)	**0.0345**

* NAFLD—non-alcoholic fatty liver disease, ** *p*-value assessed using the Chi2 test. OR—odd ratio. The bolded results indicate statistically significant differences.

## Data Availability

Marcin Kosmalski will provide data.

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
