# Peer review of "Non-Alcoholic Fatty Liver Disease Is Associated with a Decreased Catalase (CAT) Level, CT Genotypes and the T Allele of the -262 C/T CAT Polymorphism"

_cells, 2023, doi:10.3390/cells12182228_

Round 1

Reviewer 1 Report

The manuscript needs lots of work. There are references missing, significant edits needed, and simplification of the data presented is necessary. There are 12 tables and not all are necessary. Many present only a few significant data and that data tends to be similar between the different tables. Therefore, the tables could be simplified or sometimes presented as figures. More importantly, several tables should be supplementary. This way the authors could present the most important data in the manuscript and simiplify. The manuscript would be more readable. 

In addition, the Discussion needs to focus on the importance of the research in addition to the data or comparisons to other studies. This reviewer was impressed with the power compared to previous studies. This should be highlighted as should the relevance of the different parameters measured.  

English editing or at least careful editing is necessary. Is the polymorphism at -262 or -272 for CT. Mention this polymorphism is in the promoter region in the Introduction.  

English is usually fine. But the manuscript was not edited with care.  

Author Response

Thank the Reviewer’s very much for your time and valuable comments on our manuscript. The responses for all points are below. The changes were introduced into the text of manuscript, as suggested by the Reviewer.                                           

Point 1: There are 12 tables and not all are necessary. Many present only a few significant data and that data tends to be similar between the different tables. Therefore, the tables could be simplified or sometimes presented as figures. More importantly, several tables should be supplementary. This way the authors could present the most important data in the manuscript and simiplify. The manuscript would be more readable.

Response 1: We are glad for this remark. According to you remark we moved tables 6, 7, 8, 9, 10, 11 and 12 to the supplementary materials, we combine tables 6 and 8 into table S1 (supplementary materials) and table 7 and 9 into table S3 (supplementary materials). We also remove figure 2 and 3 to the supplementary materials. We hope that in the proesent form the manuscript is clearer and more readable.

Point 2: The discussion needs to focus on the importance of the research in addition to the data or comparisons to other studies. This reviewer was impressed with the power compared to previous studies. This should be highlighted as should the relevance of the different parameters measured. 

Response 2: Thank you very much for this comment. We modified the discussion according to your clues. A a result we removed unnecessary sentences and highligthed the most significant findings.

Point 3: English editing or at least careful editing is necessary. Is the polymorphism at -262 or -272 for CT. Mention this polymorphism is in the promoter region in the Introduction. 

Response 3: Indeed, we made a lot of editing errors. We read our manuscript carefully and made necessary corrections. We also added information about this polymorphism in the Introduction.

In addition, we modified our article according to the suggestions of other reviewers.

We sincerely hope that all changes introduced by us in the text will be fully satisfactory for the Reviewer.

Reviewer 2 Report

Overall Assessment:

After carefully reviewing the study by Kosmalski et al. titled "The Relationship between Catalase (CAT) Level, CAT -262 C/T Polymorphism, and Presence of Non-Alcoholic Fatty Liver Disease," I find it to be a well-written manuscript that presents the study clearly. However, I believe there is one crucial piece of data missing, which is essential to support the authors' claims regarding the management of reactive oxygen species (ROS) and oxidative stress by CAT. I recommend including measurements of ROS in the patients' serum to establish a physiological connection between the CAT data and the non-alcoholic steatohepatitis (NASH) pathology. This suggestion is based on a relevant study published in 2019, titled " Approaches and Methods to Measure Oxidative Stress in Clinical Samples: Research Applications in the Cancer Field" (Oxid Med Cell Longev. 2019; 2019: 1279250).

Minor grammatical and typographical errors were observed throughout the manuscript, which could be corrected during the revision process.

Author Response

Thank the Reviewer’s very much for your time and valuable comments on our manuscript. The responses for all points are below.

Point 1: After carefully reviewing the study by Kosmalski et al. titled "The Relationship between Catalase (CAT) Level, CAT -262 C/T Polymorphism, and Presence of Non-Alcoholic Fatty Liver Disease," I find it to be a well-written manuscript that presents the study clearly. However, I believe there is one crucial piece of data missing, which is essential to support the authors' claims regarding the management of reactive oxygen species (ROS) and oxidative stress by CAT. I recommend including measurements of ROS in the patients' serum to establish a physiological connection between the CAT data and the non-alcoholic steatohepatitis (NASH) pathology. This suggestion is based on a relevant study published in 2019, titled " Approaches and Methods to Measure Oxidative Stress in Clinical Samples: Research Applications in the Cancer Field" (Oxid Med Cell Longev. 2019; 2019: 1279250).

Response 1: Thank you for prompt comment. We agree that measurement of ROS, MDA, 8-oxoG or other maker of oxidative stress would support our data. However, we did not plan to conduct these measurement due to the fact that there are many studies showing that oxidative stress markers are elevated in NAFLD (Gonzalez-Paredes, F.J.; Mesa, G.H.; Arraez, D.M.; Reyes, R.M.; Abrante, B.; Diaz-Flores, F.;  Salido, E.; Quintero, E.; Hernández-Guerra, M. Contribution of Cyclooxygenase End Products and Oxidative Stress to Intrahepatic Endothelial Dysfunction in Early Non-Alcoholic Fatty Liver Disease. PLoS One. 2016, 11, e0156650, KöroÄŸlu, E.; Canbakan, B.; Atay, K.; Hatemi, I.; Tuncer, M.; Dobrucalı, A.; Sonsuz, A.; Gültepe, I.; Åžentürk, H.. Role of oxidative stress and insulin resistance in disease severity of non-alcoholic fatty liver disease. Turk J Gastroenterol. 2016, 27, 361-366, Daugherity, E.K.; Balmus, G.; Al Saei, A.;  Moore, E.S.; Abdallah, D.A.; Rogers, A.B.; Weiss, R.S.; Maurer, K. J. The DNA damage checkpoint protein ATM promotes hepatocellular apoptosis and fibrosis in a mouse model of non-alcoholic fatty liver disease. Cell Cycle. 2012, 11, 1918-1928). We have highlighted the role of ROS overproduction and oxidative stress in the patogenesis of NAFLD in the Introduction and Discussion. Unfortunately, we did not collect enough blood/serum samples from each participant to determine ROS level now. However, to address to you accurate remark we search throug literature and found publications that confirm increased ROS level in subjects suffering from NAFLD.

Point 2: Minor grammatical and typographical errors were observed throughout the manuscript, which could be corrected during the revision process.

Response 2: We have corrected minor grammatical errors in the article that we did not notice while writing.

In addition, we modified our article according to the suggestions of other reviewers.

Reviewer 3 Report

Very well done. Excellent design, writing and conclusions.

Author Response

We would like to thank you for kind words about our research, we are grateful for Reviewer’s time and effort to review our manuscript.

Reviewer 4 Report

The main aim of this study is to assess whether CAT level and –262C/T CAT polymorphism may be related to NAFLD.

Major comments:

1.      Many sentences should be rephrased for the sake of scientific integrity. For example, the first sentence in the introduction “Nonalcoholic fatty liver disease (NAFLD) is one of the most common causes of liver diseases”. NAFLD is a liver disease not a cause of disease.

2.      The introduction could be reorganized by focusing on the flow of information.

3.      Using ELISA to assay catalase doesn’t reflect the enzymatic activity but the presence/level of the enzyme as a protein. Enzymatic activity of catalase should be determined using appropriate method. In the figures, the authors expressed catalase in activity, but the used method doesn’t reflect that.

4.      Despite catalase is the main focus of this study, nothing related to oxidative stress or redox imbalance has been considered. The authors need to determine (A) catalase ACTIVITY and (B) oxidative stress markers.

5.      The results could be presented in a better way. Replacement of some tables with figures is recommended.

6.      There are many typos and grammatical errors that should be corrected throughout the manuscript.

 There are many typos and grammatical errors that should be corrected throughout the manuscript.

Author Response

Thank the Reviewer’s very much for your time and valuable comments on our manuscript. The responses for all points are below. The changes were introduced into the text of manuscript, as suggested by the Reviewer.

Point 1: Many sentences should be rephrased for the sake of scientific integrity. For example, the first sentence in the introduction “Nonalcoholic fatty liver disease (NAFLD) is one of the most common causes of liver diseases”. NAFLD is a liver disease not a cause of disease.

Response 1: Thank you very much for this attention. In agreement, we made the changes in the manuscript.

Point 2: The introduction could be reorganized by focusing on the flow of information.

Response 2: Thank you very much for this attention. In agreement, we made the changes in the manuscript.

Point 3: Using ELISA to assay catalase doesn’t reflect the enzymatic activity but the presence/level of the enzyme as a protein. Enzymatic activity of catalase should be determined using appropriate method. In the figures, the authors expressed catalase in activity, but the used method doesn’t reflect that.

Response 3: Thank you very much for this attention. We improved all figures by converting activity to level.

Point 4: Despite catalase is the main focus of this study, nothing related to oxidative stress or redox imbalance has been considered. The authors need to determine (A) catalase ACTIVITY and (B) oxidative stress markers.

Response 4: Thank you for prompt comment. We agree that measurement of ROS, MDA, 8-oxoG or other maker of oxidative stress would support our data. However, we did not plan to conduct these measurement due to the fact that there are many studies showing that oxidative stress markers are elevated in NAFLD (Gonzalez-Paredes, F.J.; Mesa, G.H.; Arraez, D.M.; Reyes, R.M.; Abrante, B.; Diaz-Flores, F.;  Salido, E.; Quintero, E.; Hernández-Guerra, M. Contribution of Cyclooxygenase End Products and Oxidative Stress to Intrahepatic Endothelial Dysfunction in Early Non-Alcoholic Fatty Liver Disease. PLoS One. 2016, 11, e0156650, KöroÄŸlu, E.; Canbakan, B.; Atay, K.; Hatemi, I.; Tuncer, M.; Dobrucalı, A.; Sonsuz, A.; Gültepe, I.; Åžentürk, H.. Role of oxidative stress and insulin resistance in disease severity of non-alcoholic fatty liver disease. Turk J Gastroenterol. 2016, 27, 361-366, Daugherity, E.K.; Balmus, G.; Al Saei, A.;  Moore, E.S.; Abdallah, D.A.; Rogers, A.B.; Weiss, R.S.; Maurer, K. J. The DNA damage checkpoint protein ATM promotes hepatocellular apoptosis and fibrosis in a mouse model of non-alcoholic fatty liver disease. Cell Cycle. 2012, 11, 1918-1928). We have highlighted the role of ROS overproduction and oxidative stress in the patogenesis of NAFLD in the Introduction and Discussion. ż te dodatkowe ) Unfortunately, we did not collect enough blood/serum samples from each participant to determine ROS level now.

Point 5: The results could be presented in a better way. Replacement of some tables with figures is recommended.

Response 5: We are glad for this remark. According to you remark we moved tables 6, 7, 8, 9, 10, 11 and 12 to the supplementary materials, we combine tables 6 and 8 into table S1 (supplementary materials) and table 7 and 9 into table S3 (supplementary materials). We also remove figure 2 and 3 to the supplementary materials. We hope that in the proesent form the manuscript is clearer and more readable.

Point 6: There are many typos and grammatical errors that should be corrected throughout the manuscript.

Response 6: Thank you very much for this attention. In agreement, we made the changes in the manuscript.

In addition, we modified our article according to the suggestions of other reviewers.

We sincerely hope that all changes introduced by us in the text will be fully satisfactory for the Reviewer.

Round 2

Reviewer 4 Report

The authors have addressed my comments except for the determination of CAT activity (not level) using appropriate method as well as oxidative stress markers (the reason stated by the authors is the unavailability of ssmples). These points should be seriously considered in future work.

A language editing service could be considered to correct the grammatical errors and improve the quality.